# Regulatory Roles of Noncanonical Inflammasomes in Diabetes Mellitus and Diabetes-Associated Complications

**DOI:** 10.3390/ijms26188893

**Published:** 2025-09-12

**Authors:** Young-Su Yi

**Affiliations:** Department of Life Sciences, Kyonggi University, Suwon 16227, Republic of Korea; ysyi@kgu.ac.kr; Tel.: +82-31-249-9644

**Keywords:** caspase-4, caspase-11, noncanonical inflammasome, diabetes, diabetes-associated complications

## Abstract

Inflammation is an innate immune system protecting the body from infection and injury. This process proceeds through two distinct stages: a priming phase, characterized by transcriptional activation, and a triggering phase, in which inflammasomes, cytosolic multiprotein complexes, are activated to initiate inflammatory signaling cascades. Canonical inflammasomes, the first to be identified, have been extensively implicated in the pathogenesis of diverse inflammatory disorders. In contrast, noncanonical inflammasomes have only recently been characterized, and their precise contributions to immune regulation and disease development remain incompletely defined. Diabetes mellitus (DM), simply diabetes, represents a heterogeneous group of metabolic disorders marked by chronic hyperglycemia and is associated with a broad spectrum of complications. The involvement of canonical inflammasomes in DM and its complications has been well demonstrated. More recently, however, accumulating evidence has uncovered crucial roles for noncanonical inflammasomes in the pathogenesis of DM and related complications This review comprehensively discusses current advances in understanding the regulatory functions of murine caspase-11 and human caspase-4/5 noncanonical inflammasomes in the pathogenesis of DM and diabetes-associated complications, highlighting their potential as novel therapeutic targets.

## 1. Introduction

An inflammasome is a cytosolic multiprotein complex that serves as a key platform for inducing inflammatory signaling cascades [1,2]. Canonical inflammasomes discovered earlier include those from the nucleotide-binding Oligomerization Domain (NOD)-like receptor (NLR) family, such as NLRP1, NLRP3, NLRC4, NLRP6, NLRP9, and NLRP12 as well as non-NLR absence of melanoma 2 (AIM2), interferon γ-inducible protein 16 (IFI16), and pyrin inflammasomes [1,2]. In contrast, noncanonical inflammasomes identified more recently include human caspase-4 and caspase-5, along with murine caspase-11 inflammasomes [1,2]. Inflammasomes become activated when their pattern recognition receptors (PRRs) sense specific pathogen-associated molecular patterns (PAMPs) or damage-associated molecular patterns (DAMPs) [1,2]. Although canonical and noncanonical inflammasomes respond to different sets of PAMPs and DAMPs, they ultimately trigger common downstream inflammatory signaling pathways [1,2]. Inflammasome activation triggers pyroptosis by forming gasdermin D (GSDMD) pores in the cell membranes, which also enables the concurrent release of pro-inflammatory cytokines via the GSDMD pores [1,2].

Diabetes mellitus (DM), commonly known as diabetes, is a group of common endocrine diseases characterized by sustained high blood sugar levels. Type 1 and type 2 diabetes are the two most prevalent forms of the disease. Type 1 diabetes occurs when the immune system mistakenly attacks and damages the pancreas, resulting in a lack of insulin production. Both genetic and environmental factors play a role in causing this autoimmune response [3]. In contrast, type 2 diabetes is a metabolic disorder characterized by the reduced sensitivity to insulin, leading to elevated blood glucose levels. Over time, the pancreas in individuals with type 2 diabetes may gradually lose its ability to produce insulin effectively [3]. Diabetes can lead to a variety of health complications affecting various parts of the body. These include cardiovascular disease, kidney disease, eye problems, nerve damage, foot problems, skin conditions, oral complications, and increased risk of infections [3]. Diabetes management depends on the specific type but typically includes lifestyle modifications, medication, and regular monitoring. Patients with type 1 diabetes require insulin therapy, whereas those with type 2 diabetes usually start with changes in diet and exercise along with medications, and may eventually require insulin treatment [3]. Despite available treatments and management strategies for diabetes, many patients continue to suffer from diabetes and its complications, which suggests the ongoing need for research into the molecular mechanisms underlying the pathogenesis of diabetes and its related conditions, as well as the development of more effective therapeutic approaches. A large body of previous research has demonstrated the involvement of canonical inflammasomes, particularly the NLRP3 inflammasome, in mediating inflammatory responses and contributing to various diseases [1,4,5,6,7,8]. Given the evidence linking canonical inflammasomes to metabolic dysregulation, studies have further demonstrated their contribution to the development of diabetes [9,10,11,12]. While canonical and noncanonical inflammasomes detect distinct DAMPs and PAMPs, they converge on similar downstream inflammatory signaling cascades [1,2]. Therefore, a growing body of recent evidence highlights the significance of noncanonical inflammasomes as key regulators in immune-mediated pathologies, such as rheumatic diseases, liver diseases, lung diseases, acute lethal sepsis, and inflammatory bowel disease and the diseases associated with diabetes [13,14,15,16,17,18,19,20,21,22,23,24]. This review discusses studies on the regulatory roles of noncanonical inflammasomes in the pathogenesis of diabetes and its complications, while also emphasizing their potential as therapeutic targets for disease prevention and treatment.

## 2. Noncanonical Inflammasomes

### 2.1. PRRs

PRRs are crucial components of inflammasomes, serving as sensors that detect PAMPs and DAMPs. Canonical inflammasome PRRs recognize PAMPs and DAMPs and subsequently assemble canonical inflammasomes by engaging caspase-1, either directly or via the adaptor protein ASC [1,2]. In contrast to canonical inflammasomes, noncanonical inflammasome PRRs, such as murine caspase-11 and human caspase-4 and caspase-5 can directly sense PAMPs without the involvement of caspase-1 or ASC [1,2]. Unlike different domain structures of canonical inflammasome PRRs, noncanonical inflammasome PRRs, murine caspase-11 and human caspase-4 and caspase-5, possess the same domain architecture. Each comprises three domains: an N-terminal caspase recruitment domain (CARD), a central large catalytic p20 domain, and a C-terminal small catalytic p10 domain (Figure 1A). Although they have the same structural organization, their amino acid lengths differ: caspase-4, caspase-5, and caspase-11 have 377, 434, and 373 amino acids, respectively (Figure 1A).

### 2.2. Ligands

Lipopolysaccharide (LPS) is an endotoxin derived from Gram-negative bacteria and was identified as the main PAMP detected by noncanonical inflammasomes [25,26]. Following infection with Gram-negative bacteria, LPS is internalized into host cells and subsequently recognized by caspase-4/5/11 [1,2]. LPS enters cells through endocytosis mediated by surface receptors, such as the toll-like receptor 4 (TLR4) and receptor for advanced glycation end-products (RAGE) [27]. Additionally, Gram-negative bacteria generate outer membrane vesicles (OMVs) containing LPS, which can merge with host cell membranes and be taken up by receptor-mediated endocytosis [27].

Recent studies have demonstrated that additional molecules also can act as PAMPs for noncanonical inflammasome activation. The oxidized phospholipid 1-palmitoyl-2-arachidonoyl-sn-glycero-3-phosphorylcholine (oxPAPC) has been shown to directly bind to caspase-11, initiating the caspase-11-dependent noncanonical inflammasome pathway in dendritic cells [28]. Interestingly, unlike LPS that interacts with CARD of caspase-4/5/11, oxPAPC interacts with the catalytic domain of caspase-11 and does not induce GSDMD-dependent pyroptosis [28]. This indicates that oxPAPC may not serve as a primary PAMP for noncanonical inflammasome activation and highlight the need to investigate its role in macrophages, the primary inflammatory cells.

The glycolipid lipophosphoglycan (LPG) found on the surface of *Leishmania* parasites was recently recognized as an intracellular stimulus for caspase-11 noncanonical inflammasome activation, subsequently inducing NLRP3 inflammasome activation in macrophages [29]. However, unlike LPS and oxPAPC, LPG does not interact with caspase-11 or caspase-4 and is unable to activate the noncanonical inflammasome in vitro [29], suggesting the requirement of further mechanism studies.

Secreted aspartyl proteinases (Saps), key virulence factors of *Candida albicans*, was identified as potential activators of caspase-11 noncanonical inflammasome in macrophages through a pathway dependent on type I interferon (IFN) signaling [30]. Additionally, intracellular sensors NOD and RIP2 were also shown to promote caspase-11 noncanonical inflammasome activation by regulating reactive oxygen species (ROS) balance during Gram-negative bacterial infection-induced inflammatory responses in macrophages [30]. Although these molecules exhibit the capacity to activate noncanonical inflammasomes, the underlying mechanisms remain largely unclear, necessitating further investigation.

### 2.3. Activation and Signaling Pathways

As mentioned earlier, LPS is taken up through receptor-mediated endocytosis and becomes enclosed within endosomes, from which it must be released into the cytosol for sensing by caspase-4/5/11. Guanylate-binding proteins (GBPs), a family of IFN-inducible GTPases, attach to endosomes, causing the loss of membrane integrity and membrane disruption, which eventually allows LPS to escape into the cytosol, where it can interact with caspase-4/5/11 [1,2].

Once LPS is released from endosomes, caspase-4/5/11 sense cytosolic LPS by direct binding between their CARDs and the lipid A moiety of LPS, and this interaction results in the formation of LPS-caspase-4/5/11 complexes, which subsequently oligomerize via CARD–CARD interactions to assemble caspase-4/5/11 noncanonical inflammasomes (Figure 1B) [1,2]. The caspase-4/5/11 noncanonical inflammasomes then are activated by autoproteolysis [31,32]. In murine caspase-11 and human caspase-4, cysteine residues at 254 and 258 serve as catalytic sites, facilitating autoproteolysis at aspartic acid residues at 285 and 289, respectively [31,32]. However, the autoproteolytic processing and molecular activation mechanisms of human caspase-5 are still unknown.

The activation of noncanonical inflammasomes subsequently initiate the signal transduction cascades of inflammatory responses. Upon the activation of noncanonical inflammasomes, GSDMD is proteolytically cleaved at 276 aspartic acid residue, generating two different GSDMD fragments, N-terminal (N-GSDMD) and C-terminal GSDMD (C-GSDMD) [1,2]. The N-GSDMD then migrates to the plasma membrane and produce GSDMD pores by their oligomerization, leading to cell swelling, membrane permeabilization, and eventual membrane rupture, a form of inflammatory cell death named pyroptosis [1,2]. Simultaneously, noncanonical inflammasomes promote the activation of NLRP3 canonical inflammasome by triggering potassium ion (K^+^) efflux, a crucial event for NLRP3 inflammasome activation through GSDMD pores, bacterial pore-forming toxins, the P2X7 channel, and membrane disruption [1,2]. This activation of NLRP3 inflammasome leads to the proteolytic activation of caspase-1, which in turn processes and enables the release of pro-inflammatory cytokines, such as interleukin (IL)-1β and IL-18 via GSDMD pores [1,2]. These cytokines further enhance the inflammatory response by activating additional immune cells. The activation of noncanonical inflammasomes and the noncanonical inflammasome-activated signaling pathways are described in Figure 2.

## 3. Roles of Noncanonical Inflammasomes in Diabetes and Its Complications

### 3.1. Type 2 Diabetes Mellitus (T2DM)

Type 2 diabetes mellitus (T2DM) is a chronic metabolic condition characterized by high blood glucose levels, insulin resistance, and a relative deficiency in insulin. It represents approximately 90% of all diabetes cases [33]. In T2DM, the body’s responsiveness to insulin is reduced, a condition known as insulin resistance. Initially, the body compensates by increasing insulin secretion to preserve normal blood glucose levels; however, as the disease progresses, insulin production gradually declines, ultimately leading to the development of T2DM [33].

Recent studies have highlighted the role of noncanonical inflammasomes in the pathogenesis of T2DM. Fender et al. investigated the involvement of caspase-11 noncanonical inflammasome in the context of diabetic heart disease associated with T2DM in mice [34]. Caspase-1 activation was elevated in the atria of T2DM patients [34]. Consistently, high-fat diet (HFD)-induced diabetic mice exhibited increased caspase-1 expression and activation in the left ventricle, leading to the activation of IL-1β and GSDMD [34]. In contrast, NLRP3 expression in the left ventricle remained unchanged [34]. Notably, PAR4 expression was increased, whereas caspase-1 activation was attenuated in the left ventricle of PAR4-dificient diabetic mice [34]. Gene expression of pro-caspase-11 remained unchanged; however, the level of cleaved caspase-11 was reduced in the left ventricle of mice with HFD-induced diabetes [34]. Notably, the NLRP3 inflammasome was activated, triggering inflammatory responses and pyroptosis in the same tissue [34]. These findings suggest that PAR4 drives caspase-1-dependent IL-1β production through the inflammasome-mediated pathways in the diabetic heart and that the two inflammasomes may play opposing roles in diabetic heart disease, highlighting the need for further mechanistic investigations.

Wang et al. demonstrated the relationship with and effect of oral microbiota on caspase-11 noncanonical inflammatory pathway in patients with T2DM [35]. Salivary levels of Fusobacteriota and Campilobacterota were reduced, while the relative abundance of Proteobacteria was elevated compared to healthy individuals [35]. Biochemical analyses revealed increased mRNA expression levels of caspase-4, NLRP3, caspase-1, ASC, LPS, and the pro-inflammatory cytokine IL-1β, along with a decreased expression level of insulin receptor substrate-1 in the blood of T2DM patients [35]. These findings suggest that alterations in the composition of oral microbiota may contribute to the expression and activation of the caspase-4 noncanonical and NLRP3 canonical inflammasome-mediated inflammatory pathways in T2DM.

An observational cross-sectional study by Arunachalam et al. investigated the involvement of the caspase-4 noncanonical inflammasome-mediated pyroptosis pathway in the gingival tissue of individuals with T2DM [36]. The study found increased expression of caspase-4 and GSDMD in the gingival tissue of T2DM patients [36], suggesting that noncanonical pyroptosis, driven by the caspase-4 inflammasome/GSDMD axis, plays a role in the pathogenesis of gingival tissue in T2DM.

In summary, these studies highlight the regulatory functions of murine caspase-11 and human caspase-4 noncanonical inflammasomes in the development of T2DM. However, their expression and activity varied across different tissues affected by T2DM, emphasizing the need for further investigation into the underlying mechanisms. The regulatory function of the caspase-4/11 noncanonical inflammasomes in T2DM pathogenesis is described in Figure 3.

### 3.2. Diabetic Nephropathy (DN)

Diabetic nephropathy (DN), also known as diabetic kidney disease (DKD), is a severe chronic complication of diabetes characterized by damage to the kidneys’ filtration system due to prolonged high blood glucose levels. This damage can progressively impair kidney function and may eventually lead to kidney failure [37]. DN is a major contributor to chronic kidney disease and end-stage renal disease worldwide [37].

Studies have recently demonstrated the critical involvement of noncanonical inflammasomes in the pathogenesis of DN. Chen et al. explored how gut microbiota-derived OMVs regulate tubulointerstitial inflammation in streptozotocin (STZ)-induced DN rats and HK-2 human renal tubular epithelial cells (TECs) [38]. OMVs from diabetic gut microbiota induced tubulointerstitial inflammation and kidney injury in STZ-induced DN rats by activating the caspase-11 noncanonical inflammasome [38]. Fecal bacterial extracellular vesicles (fBEVs) stimulated the caspase-4 noncanonical inflammasome, eliciting inflammatory responses, whereas silencing caspase-4 reduced this fBEV-induced inflammation in HK-2 cells [38]. These results suggest that OMVs derived from diabetic gut microbiota induce noncanonical inflammasome activation, resulting in renal inflammation and tubulointerstitial damage in DN.

Ito et al. reported that the caspase-11 noncanonical inflammasome is involved in renal injury in diabetic BTBR ob/ob mice [39]. Caspase-11 expression was elevated in the glomeruli of these diabetic mice [39], suggesting that activation of the caspase-11 noncanonical inflammasome may promote diabetic renal injury by inducing pyroptosis, thereby contributing to DN pathogenesis. However, despite the evidence of this study, the detailed molecular mechanisms by which caspase-11 noncanonical inflammasome activation and the subsequent GSDMD-mediated pyroptosis occurs in renal tissues during DN pathogenesis remain unclear and required further investigation.

Shao et al. investigated the involvement of the murine caspase-4/11 noncanonical inflammasome and GSDMD-driven pyroptosis in glomerular endothelial cells (GECs) and in a mouse model of DN induced by STZ [40]. GSDMD-driven pyroptosis was induced in mouse and human GECs treated with HG [40]. GSDMD-driven pyroptosis contributed to renal damage; however, silencing GSDMD in renal tissues mitigated kidney injury in STZ-induced DN mice [40]. Moreover, HG condition activated the caspase-11 noncanonical inflammasome and induced pyroptosis in mouse GECs, suggesting the pivotal role of caspase-11 noncanonical inflammasome activation and subsequent GSDMD-driven pyroptosis in DN pathogenesis.

Cheng et al. demonstrated the regulatory role of caspase-4/11 noncanonical inflammasomes and GSDMD-driven pyroptosis in the renal injury in podocytes and a mouse model of DN induced by HFD and STZ [41]. Caspase-11 expression was upregulated in the renal tissues of HFD/ STZ-induced DN mice, and silencing caspase-11 attenuated podocyte damage and renal inflammation in DN mice [41]. High glucose (HG) condition also upregulated caspase-4 expression, GSDMD processing, and inflammatory responses in human podocytes [41]. Similarly, GSDMD processing was induced in HFD/ STZ-induced DN mice, whereas GSDMD knockout mitigated podocyte damage in DN mice [41]. These findings imply that caspase-4/11 noncanonical inflammasomes promote GSDMD-dependent pyroptosis in podocytes, contributing to DN pathogenesis.

In summary, these findings emphasize the functional involvement of caspase-4/11 noncanonical inflammasomes in the development and progression of DN. Nonetheless, their specific contributions and underlying mechanisms in renal injury during DN pathogenesis remain largely unclear, strongly demanding the need for further research to elucidate their regulatory functions in different renal cell types during DN pathogenesis along with the associated molecular pathways. The involvement of caspase-4/11 noncanonical inflammasomes in DN pathogenesis is illustrated in Figure 4.

### 3.3. Diabetic Periodontitis (DP)

Diabetic periodontitis (DP) is a severe form of gum disease that occurs more commonly and with greater severity in people with diabetes. Individuals with advanced periodontitis have a significantly higher risk of developing diabetes compared to those with healthy gums [42]. A positive correlation exists between diabetes mellitus and periodontitis, as diabetic individuals are more prone to this condition due to compromised immune responses, delayed wound healing, and elevated glucose levels in the gingival crevicular fluid, which promote bacterial proliferation [43].

Recent studies have highlighted the pivotal roles of noncanonical inflammasomes in the pathogenesis of DP. Li et al. investigated examined the therapeutic effects of a tetrahedral framework nucleic acid (tFNA)-loaded metformin complex (TMC) on DP pathogenesis by modulating caspase-11 noncanonical inflammasome in macrophages and a DP mouse model induced by HFD and ligature [44]. TMC inhibited caspase-11 noncanonical inflammasome activation, leading to reduced pyroptosis and decreased production of pro-inflammatory cytokines in HG-treated RAW264.7 macrophages [44]. In addition, TMC suppressed inflammatory cell infiltration and osteoclast formation, thereby mitigating alveolar bone loss in HFD/ligature-induced DP mice [44]. These finding indicate that TMC may offer therapeutic benefits for DP by targeting caspase-11 noncanonical inflammasome and consequently reducing DP-related inflammation and bone resorption.

Arunachalam et al. reported an observational cross-sectional study to investigate the role of pyroptosis triggered by caspase-4 noncanonical inflammasome activation in the gingival tissues of patients with DP and T2DM [36]. Caspase-4 and GSDMD expression levels were elevated in the gingiva of these patients [36]. Also, caspase-4 and GSDMD expression positively correlated with periodontal and diabetic parameters [36]. These results suggest that GSDMD-driven pyroptosis induced by caspase-4 noncanonical inflammasome activation may critically contribute to the pathogenesis of DP in individuals with T2DM.

In summary, these findings provide critical evidence that caspase-4/11 noncanonical inflammasomes are associated with DP by promoting pyroptosis and inflammatory responses. However, their regulatory roles in the pathogenesis of diabetes-associated DP, as well as the underlying molecular mechanisms are still poorly understood, highlighting the demand for further studies to elucidate their regulatory roles and mechanisms in the pathogenesis of diabetes-associated DP. The regulatory functions of caspase-4/11 noncanonical inflammasomes in diabetes-associated DP pathogenesis are depicted in Figure 5.

### 3.4. Other Diabetes-Associated Complications

Enteric neuronal degeneration describes the progressive damage and loss of enteric neurons, the nerve cells within the enteric nervous system [45,46]. Degeneration of these neurons can lead to various gastrointestinal problems and is associated with both aging and neurodegenerative diseases [45,46]. In obesity and diabetes, neurodegeneration within the enteric nervous system can be accelerated, a condition termed diabetic enteric neuropathy (DEN) [47]. DEN results in structural and functional damage to the enteric nervous system, contributing to gastrointestinal issues, such as delayed or rapid gastric emptying, diarrhea, and constipation [48,49]. As a major complication of diabetes, understanding the mechanisms of DEN and exploring potential therapies to protect or regenerate the enteric nervous system are crucial for managing gastrointestinal complications in diabetic patients.

A recent study demonstrated the regulatory function of the noncanonical inflammasome in DEN pathogenesis. Ye et al. showed that caspase-11 noncanonical inflammasome-mediated pyroptosis contributes to DEN pathogenesis in mice with Western diet (WD)-induced enteric nitrergic neuronal degeneration [50]. Elevated pyroptosis levels in the myenteric ganglia were detected in overweight and obese human subjects [50]. An increase in myenteric neuronal pyroptosis accompanied by delayed colonic transit and reduced colonic relaxation responses to electric field stimulation were exhibited in mice with WD-induced enteric nitrergic neuronal degeneration [50]. Moreover, WD promoted pyroptosis of myenteric nitrergic neurons and impaired colonic motility, whereas caspase-11 deficiency prevented both nitrergic myenteric neuronal pyroptosis and colonic dysmotility in mice with enteric nitrergic neuronal degeneration [50]. In vitro study showed that palmitate and LPS induced pyroptosis in nitrergic enteric neurons [50]. These results indicate that the caspase-11 noncanonical inflammasome-triggered pyroptotic pathway contributes to WD-induced myenteric nitrergic neuronal degeneration and colonic dysmotility, highlighting potential therapeutic targets for enteric neuropathy.

In diabetes, particularly when kidney function is impaired, excess fluid can accumulate in the body, resulting in hypervolemia [51]. This condition may arise from multiple factors, including impaired glycemic control, hormonal imbalances, and renal dysfunction [51]. Hypervolemic hemodialysis (HD) refers to dialysis performed in patients with this hypervolemia. Managing HD in individuals with diabetes can be especially complex, as these patients may experience significant blood glucose fluctuations due to interactions between dialysis, insulin dynamics, and counter-regulatory hormones, potentially leading to both hypoglycemia and hyperglycemia [52,53]. Consequently, diabetic hypervolemic hemodialysis (DHD) represents a crucial component of care in diabetic patients with renal failure, necessitating an integrated approach that manages both fluid status and blood glucose levels [52,53].

A study highlighted the regulatory involvement of the noncanonical inflammasome in diabetic HD (DHD) pathogenesis. In a cross-sectional analysis, Ulrich et al. examined the involvement of the caspase-4 noncanonical inflammasome in inflammatory responses among DHD patients and found that the activity of caspase-4 noncanonical inflammasome was significantly elevated in DHD patient serum [54]. Caspase-4 noncanonical inflammasome is activated by intracellular LPS [1,2], thus, the elevated caspase-4 noncanonical inflammasome activity found in DHD patients may be an indication of a higher intracellular bacterial load in these patients Interestingly, serum endotoxin levels were not elevated [54], suggesting the need for further studies to clarify the relationship between serum endotoxin concentrations and caspase-4 noncanonical inflammasome activity in DHD. This study suggests the involvement of caspase-4 noncanonical inflammasome as a contributor to DHD pathogenesis, though its exact role and underlying mechanisms remain to be fully elucidated.

In summary, these findings suggest that noncanonical inflammasomes contribute to the pathogenesis of diabetes-associated complications, such as DEN and DHD. However, the molecular mechanisms of caspase-11 noncanonical inflammasome-triggered pyroptosis in enteric neuronal pyroptosis in obese subjects or diabetes patients are still largely unknown. Furthermore, only increased expression of caspase-4 was observed in DHD pathogenesis, which requires demonstrating the regulatory functions of caspase-4 noncanonical inflammasome along with the underlying mechanisms during the pathogenesis of DHD. The regulatory functions of noncanonical inflammasomes in the pathogenesis of DEN and DHD are illustrated in Figure 6.

## 4. Conclusions and Perspectives

This study provides a comprehensive synthesis of recent advances delineating the regulatory functions of murine caspase-11 and human caspase-4 noncanonical inflammasomes in the pathogenesis of diabetes mellitus and its associated complications, including T2DM, DN, DP, DHD, and DEN. Emerging evidence from relevant animal models and diverse disease-associated cellular models, as summarized in Table 1, has revealed novel molecular and cellular mechanisms underlying these diseases. Although the roles of noncanonical inflammasomes may vary depending on the specific diabetic complication, their activation is consistently associated with the diseases through triggering GSDMD-mediated pyroptosis and inflammatory responses at disease lesions, which provides the therapeutic potential of selectively targeting noncanonical inflammasomes for diabetes and its complications.

Although noncanonical inflammasomes have emerged as regulators in diabetes and its complications, canonical inflammasomes, particularly NLRP3, remain better established as central mediators through their broad regulation of inflammatory signaling pathways across diverse cell types [55,56,57,58]. In contrast, noncanonical inflammasomes have been implicated in only a subset of diabetic complications, with their mechanisms still poorly understood. Unlike canonical inflammasomes, which respond to a wide range of PAMPs and DAMPs, noncanonical inflammasomes are activated exclusively by intracellular LPS, suggesting a more selective therapeutic potential in diabetes associated with Gram-negative bacterial infections. Notably, a growing body of evidence indicates that noncanonical inflammasomes also exert crucial functions in numerous pathological conditions characterized by sterile inflammation [22,24,59,60,61,62,63,64,65,66]. This strongly suggests that noncanonical inflammasomes may have broader regulatory roles across a wide spectrum of human diseases, including both sterile and non-sterile inflammatory conditions in diabetes and its complications, emphasizing the need for deeper investigation into their functions across both sterile and non-sterile inflammatory settings in diabetes and beyond.

Despite substantial evidence supporting the regulatory roles of noncanonical inflammasomes in diabetes and its complications, most studies discussed in this review have primarily focused on murine caspase-11 using mouse and cell models. Although murine caspase-11 is the functional ortholog of human caspase-4/5, they may differ in the following aspects. 1) Murine possesses only a single caspase-11, whereas humans have both caspase-4 and caspase-5, which may differ in their expression patterns (normal vs. pathological conditions), activation requirements, and relative contributions to disease pathogenesis. Human data are often caspase-4-focused, with the role of caspase-5 being less well defined. Therefore, direct generalization of murine caspase-11 findings to human caspase-4/5 as a whole may have the risk of either underestimating or overestimating their relevance. 2) The structural chemistry of LPS lipid A may be different between murine caspase-11 and human caspase-4/5. Caspase-4 can recognize certain under-acylated forms of lipid A (e.g., those from *Francisella* species) more effectively than caspase-11, leading to pronounced differences in human–mouse responses across pathogens [67]. Consequently, findings related to specific pathogens or commensals should be reinterpreted in context of these interspecies differences. 3) The basal expression levels of human caspase-4/5 and their dependence on IFN-mediated priming vary across tissues and disease contexts, which makes it difficult to directly extrapolate responses observed in murine tissues to specific human organs (e.g., brain, heart, kidney) [68]. 4) Standardized assays to determine active caspase-4/5 and GSDMD in blood and other body fluids, as well as selective pharmacological agents, particularly caspase-4/5-specific inhibitors, are not yet available for clinical use [69,70]. Therefore, findings from murine caspase-11 studies demonstrating a simple intervention and improvement in animal disease models cannot be readily translated to patient trials. 5) Experimental LPS administration for acute systemic exposure differs from the exposure patterns observed in many chronic or localized human pathologies, such as chronic inflammatory diseases [71]. Consequently, results from murine endotoxin models may not directly translate and should be interpreted carefully. These emphasize the need for translational studies and clinical trials to confirm the functions of murine caspase-11 noncanonical inflammasome in human patients with diabetes and its complications. In addition, studies of human caspase-4 noncanonical inflammasome have largely been restricted to in vitro studies with human cells and only rarely extended to patient samples, highlighting the importance of further elucidating the contributions of human caspase-4/5 noncanonical inflammasomes in the context of diabetes and diabetes-associated pathologies. Furthermore, the molecular mechanisms by which noncanonical inflammasomes drive the pathogenesis of diabetes and its complications, as well as their functional crosstalk with other molecular pathways in these contexts, remain poorly defined. This highlights the need for future investigations aimed at identifying and validating novel molecules that functionally interplay with noncanonical inflammasomes and delineating the mechanistic basis of their functional interplay in diabetic pathophysiology. As described above, the activation of the caspase-11 noncanonical inflammasome induces GSDMD pore formation, leading to K^+^ efflux through these pores, a critical trigger for NLRP3 inflammasome activation [1,2]. Conversely, the NLRP3 inflammasome can promote caspase-11 noncanonical inflammasome activation via coordinated interactions with bacterial PAMPs during infection [72]. These findings indicate that the caspase-11 noncanonical inflammasome functionally interplays with the NLRP3 canonical inflammasome in inflammatory and infectious contexts, rather than operating independently. Nevertheless, direct evidence for this functional interplay in diabetes and diabetes-associated complications remains lacking and requires further investigation. Finally, the development of potential therapeutics that selectively target noncanonical inflammasomes, together with translational studies to evaluate their efficacy in patients with diabetes and its complications, is of critical importance.

In conclusion, murine caspase-11 and human caspase-4 noncanonical inflammasomes are pivotal contributors to the pathogenesis of diabetes and diabetes-associated complications, primarily by promoting tissue damage through GSDMD-mediated pyroptosis and inflammatory responses at disease sites via functional interactions with other cellular molecules. Defining the regulatory functions and the underlying molecular mechanistic pathways of noncanonical inflammasomes in association with diabetes and its complications will be critical for the development of targeted therapies and the advancement of translational studies in the patients suffering from diabetes and diabetes-associated complications.

## Figures and Tables

**Figure 1 ijms-26-08893-f001:**
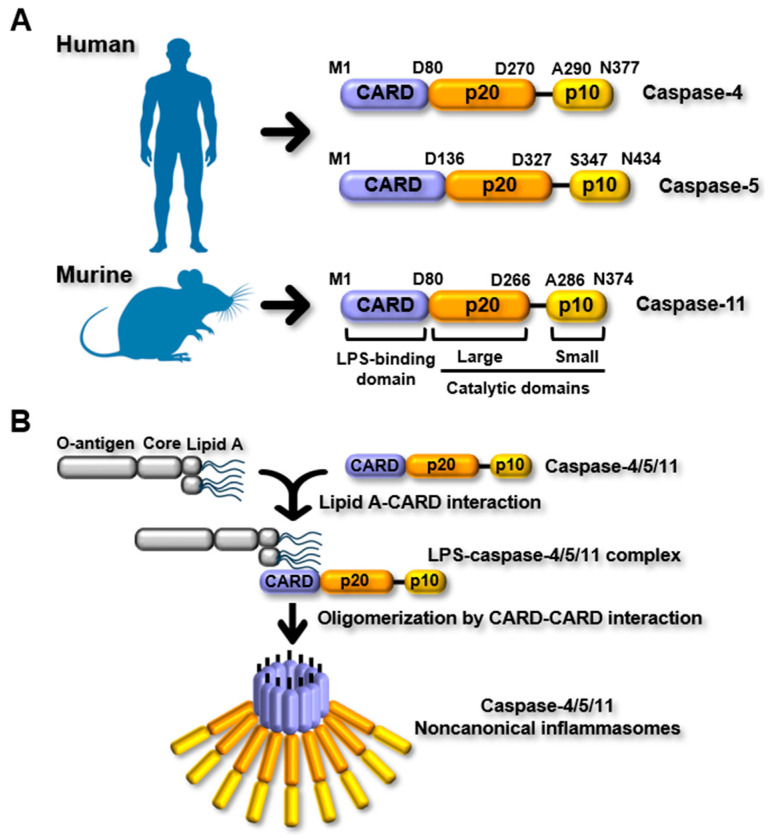
Noncanonical inflammasomes (**A**) PRRs of noncanonical inflammasomes. Human caspase-4, caspase-5, and murine caspase-11 have highly conserved structural organization, each comprising an N-terminal CARD, an LPS-binding domain, and two catalytic domains, the large p20 and small p10 domains, at the C-terminus. The polypeptide lengths of caspase-4, caspase-5, and caspase-11 are 377, 434, and 373 amino acids, respectively. (**B**) Ligand recognition and activation of noncanonical inflammasomes. Activation of noncanonical inflammasomes is initiated when caspase-4, caspase-5, or caspase-11 directly recognize LPS in the cytosol. This recognition is mediated by the CARDs of the caspases, which binds to the lipid A portion of LPS, resulting in the formation of LPS–caspase-4/5/11 complexes. These complexes subsequently undergo oligomerization through CARD–CARD interactions, thereby assembling the caspase-4/5/11 noncanonical inflammasomes.

**Figure 2 ijms-26-08893-f002:**
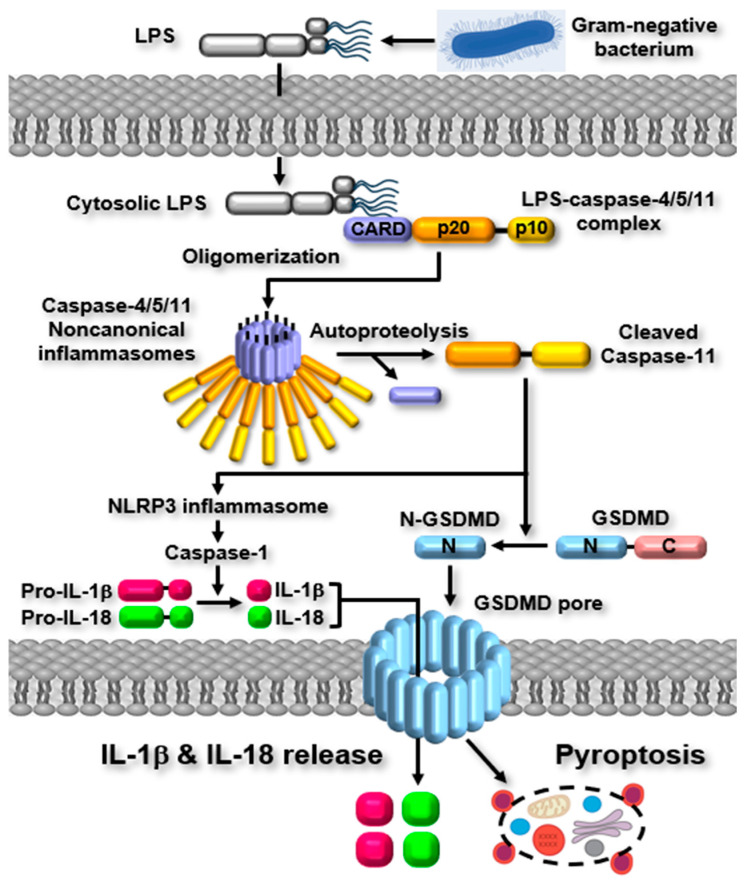
Noncanonical inflammasome-activated inflammatory signaling pathways. Caspase-4/5/11 noncanonical inflammasomes undergo autoproteolytic cleavage to generate their active forms, which subsequently process GSDMD. The cleavage of GSDMD results in pore formation within the plasma membrane, thereby driving pyroptotic cell death. In parallel, active caspase-4/5/11 noncanonical inflammasomes engage the NLRP3 inflammasome, leading to proteolytic activation of caspase-1. Activated caspase-1, in turn, mediates the proteolytic maturation of the pro-inflammatory cytokines IL-1β and IL-18, which are released through GSDMD pores.

**Figure 3 ijms-26-08893-f003:**
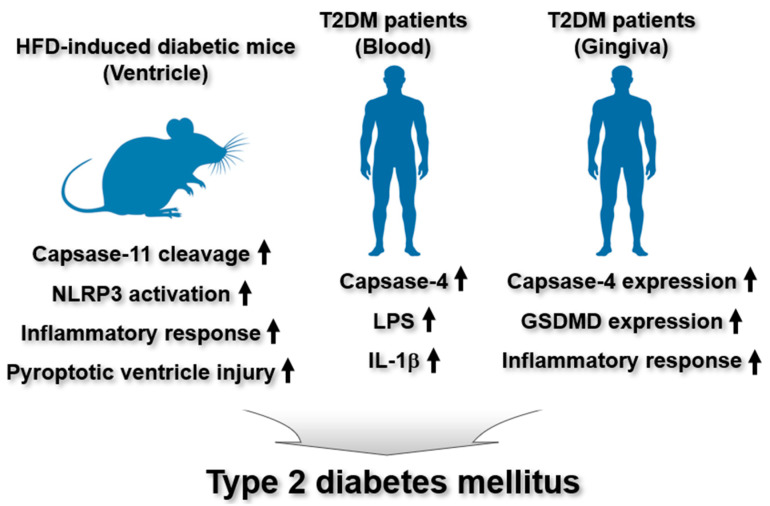
Regulatory roles of noncanonical inflammasomes in T2DM. In HFD-induced diabetic mice, ventricular tissues exhibit enhanced caspase-11 autoproteolysis accompanied by elevated caspase-11 noncanonical inflammasome–dependent inflammatory responses and pyroptosis. In patients with T2DM, serum levels of caspase-4, LPS, and IL-1β are markedly increased. Consistently, gingival tissues from T2DM patients show upregulated expression of caspase-4 and GSDMD, together with caspase-4 noncanonical inflammasome-activated inflammatory responses.

**Figure 4 ijms-26-08893-f004:**
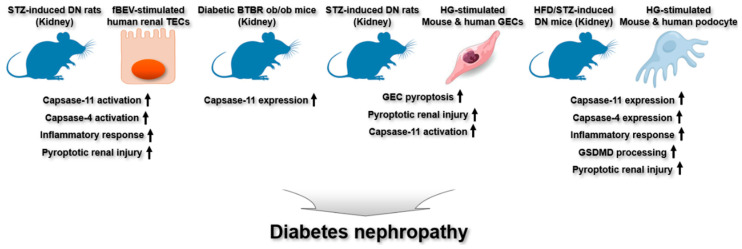
Regulatory roles of noncanonical inflammasomes in DN. In STZ- or HFD/STZ-induced models of DN in mice and rats, renal tissues show caspase-11 noncanonical inflammasome activation, accompanied by caspase-11 noncanonical inflammasome-mediated inflammatory responses and pyroptotic injury. Similarly, caspase-11 expression is elevated in the kidneys of diabetic BTBR ob/ob mice. Caspase-4 and caspase-11 noncanonical inflammasomes are activated in fBEV-stimulated human TECs, as well as in HG-stimulated mouse and human GECs and podocytes, leading to inflammatory responses, GSDMD processing, and pyroptosis.

**Figure 5 ijms-26-08893-f005:**
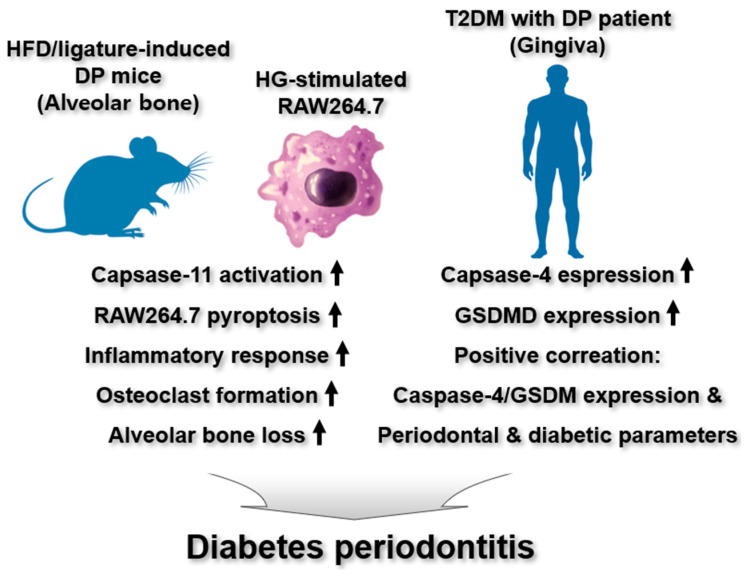
Regulatory roles of noncanonical inflammasomes in DP. In HG-stimulated mouse RAW264.7 macrophages, activation of the caspase-11 noncanonical inflammasome drives inflammatory responses and pyroptosis. In an HFD/ligature-induced mouse model of DP, caspase-11 noncanonical inflammasome is activated, promoting osteoclastogenesis and alveolar bone resorption. Gingival tissues from patients with T2DM and DP show elevated expression of caspase-4 and GSDMD, with their expression levels positively correlating with both periodontal disease severity and diabetic clinical parameters.

**Figure 6 ijms-26-08893-f006:**
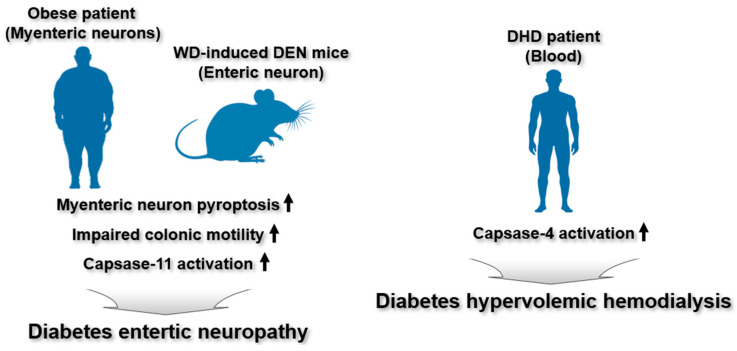
Regulatory roles of noncanonical inflammasomes in DEN and DHD. In overweight and obese patients, nitrergic enteric neurons exhibit increased susceptibility to pyroptotic cell death. In a WD-induced mouse model of DEN, activation of the caspase-11 noncanonical inflammasome drives pyroptosis of myenteric neurons, resulting in impaired colonic motility. Caspase-4 noncanonical inflammasome activity is elevated in the serum of DHD patients.

**Table 1 ijms-26-08893-t001:** Regulatory roles of noncanonical inflammasomes in diabetes and diabetes-associated complications.

Diseases	Caspase	Roles	Models	Ref.
T2DM	Caspase-11	Caspase-1 activation was elevated in the atria of T2DM patients HFD-induced diabetic mice exhibited increased caspase-1 expression and activation in the left ventricle, leading to the activation of IL-1β and GSDMD NLRP3 expression in the left ventricle remained unchanged PAR4 expression was increased, whereas caspase-1 activation was attenuated in the left ventricle of PAR4-dificient diabetic mice Pro-caspase-11 expression remained unchanged, but cleaved caspase-11 level was reduced in left ventricle of HFD-induced diabetic mice NLRP3 inflammasome was activated, triggering inflammatory responses and pyroptosis in left ventricle of HFD-induced diabetic mice	T2DM patients HFD-induced diabetic mice	[34]
	Caspase-4	Salivary levels of Fusobacteriota and Campilobacterota were reduced, while the relative abundance of Proteobacteria was elevated compared to healthy individuals mRNA expression levels of caspase-4, NLRP3, caspase-1, ASC, LPS, and the pro-inflammatory cytokine IL-1β was increased in the blood of T2DM patients mRNA expression level of insulin receptor substrate-1 was decreased in the blood of T2DM patients	T2DM patients	[35]
	Caspase-4	Expression levels of caspase-4 and GSDMD were elevated in gingival tissue of T2DM patients	T2DM patients	[36]
DN	Caspase-11	OMVs from diabetic gut microbiota activated caspase-11 noncanonical inflammasome in STZ-induced DN rats OMVs induced tubulointerstitial inflammation and kidney injury in STZ-induced DN rats fBEVs stimulated caspase-4 noncanonical inflammasome, eliciting inflammatory responses in HK-2 cells Caspase-4 silencing reduced fBEV-induced inflammation in HK-2 cells	STZ-induced DN rats HK-2 human renal TECs	[38]
	Caspase-11	Caspase-11 expression was elevated in glomeruli of diabetic BTBR ob/ob mice	Diabetic BTBR ob/ob mice	[39]
	Caspase-4/11	GSDMD-driven pyroptosis was induced in HG-treated mouse and human GECs GSDMD-driven pyroptosis contributed to renal damage; however, silencing GSDMD in renal tissues mitigated kidney injury in STZ-induced DN mice HG conditions activated caspase-11 noncanonical inflammasome and induced pyroptosis in mouse GECs	STZ-induced DN mice Mouse and human GECs	[40]
	Caspase-4/11	Caspase-11 expression was upregulated in renal tissues of HFD/STZ-induced DN mice Silencing caspase-11 attenuated podocyte damage and renal inflammation in HFD/STZ-induced DN mice HG condition upregulated caspase-4 expression, GSDMD processing, and inflammatory responses in human podocytes GSDMD processing was induced in HFD/STZ-induced DN mice, whereas GSDMD knockout mitigated podocyte damage in HFD/STZ-induced DN mice	HFD/STZ-induced DN mice Mouse and human podocytes	[41]
DP	Caspase-11	TMC inhibited caspase-11 noncanonical inflammasome activation, leading to reduced pyroptosis and decreased production of pro-inflammatory cytokines in HG-treated RAW264.7 macrophages TMC suppressed inflammatory cell infiltration and osteoclast formation, thereby mitigating alveolar bone loss in HFD/ligature-induced DP mice	HFD/ligature-induced DP mice RAW264.7	[44]
	Caspase-4	Caspase-4 and GSDMD expression levels were elevated in the gingiva of T2DM/DP patients Caspase-4 and GSDMD expression positively correlated with periodontal and diabetic parameters in of T2DM/DP patients	T2DM/DP patients	[36]
DEN	Caspase-11	Elevated pyroptosis levels in myenteric ganglia were detected in overweight and obese human subjects An increase in myenteric neuronal pyroptosis accompanied by delayed colonic transit and reduced colonic relaxation responses to electric field stimulation were exhibited in mice with WD-induced enteric nitrergic neuronal degeneration WD promoted pyroptosis of myenteric nitrergic neurons and impaired colonic motility in mice with enteric nitrergic neuronal degeneration Caspase-11 deficiency prevented both nitrergic myenteric neuronal pyroptosis and colonic dysmotility in mice with enteric nitrergic neuronal degeneration Palmitate and LPS induced pyroptosis in nitrergic enteric neurons	Obese patients WD-induced enteric nitrergic neuronal degeneration mice	[50]
DHD	Caspase-4	Caspase-4 noncanonical inflammasome activity was significantly elevated in DHD patient serum	DHD patients	[54]

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
