# Peer review of "Regulatory Roles of Noncanonical Inflammasomes in Diabetes Mellitus and Diabetes-Associated Complications"

_ijms, 2025, doi:10.3390/ijms26188893_

Round 1
Reviewer 1 Report
Comments and Suggestions for Authors
The first part of the review examines the structure of noncanonical inflammasomes, their differences from canonical ones, the ligands with which they interact, and the molecular mechanisms of their activation. The second part provides information on the role of noncanonical inflammasomes in Type 2 diabetes mellitus and diabetic complications (diabetic nephropathy, diabetic peritonitis, diabetic enteric neuropathy, diabetic hypervolemic hemodialysis). It is emphasized that the molecular mechanisms by which inflammasomes participate in the pathogenesis of diabetes remain unexplored. The review is of interest to physicians and researchers studying diabetes and can be recommended for publication. However, it seems that the content of the review is better reflected in the following title: Regulatory roles of noncanonical inflammasomes in diabetes mellitus 2 and diabetes-associated complications

Author Response
The first part of the review examines the structure of noncanonical inflammasomes, their differences from canonical ones, the ligands with which they interact, and the molecular mechanisms of their activation. The second part provides information on the role of noncanonical inflammasomes in Type 2 diabetes mellitus and diabetic complications (diabetic nephropathy, diabetic peritonitis, diabetic enteric neuropathy, diabetic hypervolemic hemodialysis). It is emphasized that the molecular mechanisms by which inflammasomes participate in the pathogenesis of diabetes remain unexplored. The review is of interest to physicians and researchers studying diabetes and can be recommended for publication. However, it seems that the content of the review is better reflected in the following title: Regulatory roles of noncanonical inflammasomes in diabetes mellitus 2 and diabetes-associated complications
- I sincerely appreciate your valuable suggestion and insightful comment. This review primarily focuses on type 2 diabetes mellitus, a metabolic disorder, and its associated complications. While inflammasomes are central to inflammatory and autoimmune diseases, the review does not address the role of noncanonical inflammasomes in type 1 diabetes mellitus, which is an inflammatory autoimmune condition. However, this review does cover diabetes-associated inflammatory conditions, such as diabetic nephropathy and diabetic periodontitis. Given this scope, I was wondering whether the original title remains appropriate.

Reviewer 2 Report
Comments and Suggestions for Authors
Reviewer comments and suggestions
This review discusses current advances in understanding the regulatory functions of murine caspase-11 and human caspase-4/5 noncanonical inflammasomes in the pathogenesis of DM and diabetes-associated complications, highlighting their potential as novel therapeutic targets.
Decision: Minor revision is needed.
Based on my view, below are the comments that need to be incorporated in the revised version of the manuscript.
- Please describe the noncanonical inflammasomes and canonical for the lay man, I did not see any explanation regarding this, also the importance or relevant in this case of diabetes
- Line 44, please avoid adding more references at a single place rather than you could discuss that paper in few lines and cite
- The authors need to reduce the plagiarism lower than 15%
- How about retinopathy relevant to the complications discussed in related to diabetes? please include as well
- Other forms that can be cardiovascular diseases that are also included in the manuscript.
- The authors can cite several other references that can be included in the manuscript and discussed
- Table 1 should be well prepared based on there study objective.
Author Response
Reviewer 2:
This review discusses current advances in understanding the regulatory functions of murine caspase-11 and human caspase-4/5 noncanonical inflammasomes in the pathogenesis of DM and diabetes-associated complications, highlighting their potential as novel therapeutic targets.
Decision: Minor revision is needed.
Based on my view, below are the comments that need to be incorporated in the revised version of the manuscript.
1. Please describe the noncanonical inflammasomes and canonical for the lay man, I did not see any explanation regarding this, also the importance or relevant in this case of diabetes
- Thank you for your comments. The canonical and noncanonical inflammasomes have been described in the Introduction section as follows (line 27-33):
“Canonical inflammasomes discovered earlier include those from the nucleotide-binding Oligomerization Domain (NOD)-like receptor (NLR) family, such as NLRP1, NLRP3, NLRC4, NLRP6, NLRP9, and NLRP12 as well as non-NLR absence of melanoma 2 (AIM2), interferon γ-inducible protein 16 (IFI16), and pyrin inflammasomes. In contrast, noncanonical inflammasomes identified more recently include human caspase-4 and caspase-5, along with murine caspase-11 inflammasomes”.
- Additionally, the importance or relevance of canonical and noncanonical inflammasomes in diabetes have been described in the Introduction section as follows (line 60-69):
“A large body of previous research has demonstrated the involvement of canonical inflammasomes, particularly the NLRP3 inflammasome, in mediating inflammatory responses and contributing to various diseases. Given the evidence linking canonical inflammasomes to metabolic dysregulation, studies have further demonstrated their contribution to the development of diabetes. While canonical and noncanonical inflammasomes detect distinct DAMPs and PAMPs, they converge on similar downstream inflammatory signaling cascades. Therefore, a growing body of recent evidence highlights the significance of noncanonical inflammasomes as key regulators in immune-mediated pathologies, such as rheumatic diseases, liver diseases, lung diseases, acute lethal sepsis, and inflammatory bowel disease and the diseases associated with diabetes.”
2. Line 44, please avoid adding more references at a single place rather than you could discuss that paper in few lines and cite
- Thank you for your comment. These references are the similar studies demonstrating the regulatory roles of noncanonical inflammasomes in the pathogenesis of several immune-mediated pathologies, therefore, several references were added in this description. As per your advice, the types of immune-mediated pathologies associated with the involvement of noncanonical inflammasomes have been newly added in this sentence.
3. The authors need to reduce the plagiarism lower than 15%
- Thank you for your comment, and I understand your concern regarding the similarity and plagiarism of the manuscript. I tried to do my best to reduce the similarity as low as possible by rephrasing every single sentence and avoid plagiarism, but it is still higher than 15%. Please understand it, and I will work with the publisher in this regard once this article is accepted.
4. How about retinopathy relevant to the complications discussed in related to diabetes? please include as well
- I sincerely appreciate your valuable suggestion and insightful comment. This review primarily addresses the regulatory functions of “noncanonical inflammasomes” in diabetes and its related complications. To date, however, no studies have investigated the involvement of noncanonical inflammasomes in diabetes-associated retinopathy. Consequently, this condition is not covered in the current review.
5. Other forms that can be cardiovascular diseases that are also included in the manuscript.
- I sincerely appreciate your valuable suggestion and insightful comment. As answered in comment 4, This review addresses the roles of “noncanonical inflammasomes” in diabetes and its related complications, and to date, no studies have investigated the involvement of noncanonical inflammasomes in diabetes-associated cardiovascular diseases, such as atherosclerosis, heart failure, coronary artery disease, cardiomyopathy etc... Consequently, these cardiovascular diseases are not covered in this review.
6. Table 1 should be well prepared based on there study objective.
- Thank you for your comment. Table 1 has been objectively prepared by only summarizing the results of all studies discussed in this review.

Reviewer 3 Report
Comments and Suggestions for Authors
The review addresses an important and underexplored aspect of diabetes pathophysiology, namely the role of noncanonical inflammasomes. The manuscript succeeds in compiling a wide range of studies, but several areas like my below comments, would benefit from clearer technical framing and closer alignment
1.Important diabetic complications such as retinopathy, peripheral neuropathy, cardiomyopathy, and NAFLD/NASH are not discussed, while less widely recognized categories (e.g., “diabetic hypervolemic hemodialysis”) are included despite limited supporting data. The rationale for prioritizing certain conditions over more established complications should be explained, ideally with reference to the strength of mechanistic evidence available.
2.Several human studies cited (for example, in DP and DHD) are largely based on expression or activity correlations without mechanistic intervention. In these cases, statements like “contributes to onset and progression” should be moderated to “is associated with” unless there is clear causal evidence (e.g., loss- or gain-of-function with downstream rescue). To add clarity, consider including a short “Strength of Evidence” note for each disease subsection (e.g., animal mechanistic, human mechanistic, or human associative only).
3.The DP section is labeled as “Figure 4,” but Figure 4 has already been used for DN. Please renumber figures sequentially and ensure that all in-text references correspond to the corrected numbering.
4.There are a number of typographical errors and word breaks (for example, “inflammaome,” “noncanon- ical,” “inflammaome-activated”). A careful copyedit is needed to correct spelling and maintain consistency in abbreviations (e.g., HG vs. “high glucose”). Please also check that each figure legend accurately matches its figure and that all abbreviations are defined at first mention.
5.The cardiac findings attributed to Fender et al. are presented as caspase-11 specific, yet the original study is centered on PAR4-driven NLRP3 signaling. Similarly, the Wang et al. paper on oral microbiota is primarily focused on NLRP3, though the review highlights serum “caspase-4” increases. The primary data should be revisited to confirm exactly what was measured (protein, activity, mRNA; and which caspase) and the text and Table 1 should be revised accordingly.
6.Much of the review relies on murine caspase-11 models, with conclusions then extended to human diabetes. Caspase-4/5 in humans, however, may differ from murine caspase-11 in ligand recognition, expression patterns, and disease relevance. It would strengthen the review if the authors clearly outlined where murine findings can reasonably be extrapolated to humans and where key translational gaps remain.
7.Both NLRP3 and caspase-4/5/11 inflammasomes are discussed, but it is not clear whether noncanonical activation acts as an independent driver of pathology or mainly serves to amplify canonical NLRP3 signaling. Authors could clarify this, as it will would improve the mechanistic framework and help identify which pathway is the more promising therapeutic target.
8.At relevant place in manuscript, the manuscript could distinguish more explicitly between noncanonical inflammasome activation in metabolic endotoxemia (arising from microbial LPS leakage) and in sterile injury contexts (such as high glucose or ROS). Highlighting these differences would provide sharper mechanistic insight and point to areas where further research is needed.
Author Response
Reviewer 3:
The review addresses an important and underexplored aspect of diabetes pathophysiology, namely the role of noncanonical inflammasomes. The manuscript succeeds in compiling a wide range of studies, but several areas like my below comments, would benefit from clearer technical framing and closer alignment
1. Important diabetic complications such as retinopathy, peripheral neuropathy, cardiomyopathy, and NAFLD/NASH are not discussed, while less widely recognized categories (e.g., “diabetic hypervolemic hemodialysis”) are included despite limited supporting data. The rationale for prioritizing certain conditions over more established complications should be explained, ideally with reference to the strength of mechanistic evidence available.
- I sincerely appreciate your valuable suggestion and insightful comment. This review primarily addresses the regulatory functions of “noncanonical inflammasomes” in diabetes and its related complications. To date, however, no studies have investigated the involvement of noncanonical inflammasomes in the diseases you listed above (retinopathy, peripheral neuropathy, cardiomyopathy, and NAFLD/NASH). Consequently, these diseases are not covered in the current review.
2. Several human studies cited (for example, in DP and DHD) are largely based on expression or activity correlations without mechanistic intervention. In these cases, statements like “contributes to onset and progression” should be moderated to “is associated with” unless there is clear causal evidence (e.g., loss- or gain-of-function with downstream rescue). To add clarity, consider including a short “Strength of Evidence” note for each disease subsection (e.g., animal mechanistic, human mechanistic, or human associative only).
- Thank you for your advice. As per your advice, “contributes to onset and progression” have been corrected to “is associated with” in the entire manuscript With respect to including a brief “Strength of Evidence” note for each disease subsection, as you mentioned, most available studies merely reported the functional involvement of noncanonical inflammasomes in these diseases, without elucidating the underlying mechanisms in either animal models or human patients. Therefore, the extent to which such notes can be provided for each subsection is limited.
3. The DP section is labeled as “Figure 4,” but Figure 4 has already been used for DN. Please renumber figures sequentially and ensure that all in-text references correspond to the corrected numbering.
- Thank you for your advice, and this is my mistake. Figure 4 in the DP section has been corrected to Figure 5.
4. There are a number of typographical errors and word breaks (for example, “inflammaome,” “noncanon- ical,” “inflammaome-activated”). A careful copyedit is needed to correct spelling and maintain consistency in abbreviations (e.g., HG vs. “high glucose”). Please also check that each figure legend accurately matches its figure and that all abbreviations are defined at first mention.
- Thank you for your comments. All typographical errors and word breaks have been corrected throughout the manuscript. All abbreviations are defined upon first appearance and used consistently thereafter. Each figure legend has been carefully aligned with its corresponding figure.
5. The cardiac findings attributed to Fender et al. are presented as caspase-11 specific, yet the original study is centered on PAR4-driven NLRP3 signaling. Similarly, the Wang et al. paper on oral microbiota is primarily focused on NLRP3, though the review highlights serum “caspase-4” increases. The primary data should be revisited to confirm exactly what was measured (protein, activity, mRNA; and which caspase) and the text and Table 1 should be revised accordingly.
- Thank you for your comments. Since this review focused on the regulatory roles of noncanonical inflammasomes, such as murine caspase-11 and human caspase-4 noncanonical inflammasomes in diabetes and its associated complications, the results only related to these noncanonical inflammasomes were discussed in this review. As per your advice, original results including NLRP3 canonical inflammasome-related results have been included in the manuscript (1. Fender et al. study: Line 187- 192 & 195-199, 2. Wang et al. study: Line 203-209) and Table 1 contents (results) have been accordingly revised.
6. Much of the review relies on murine caspase-11 models, with conclusions then extended to human diabetes. Caspase-4/5 in humans, however, may differ from murine caspase-11 in ligand recognition, expression patterns, and disease relevance. It would strengthen the review if the authors clearly outlined where murine findings can reasonably be extrapolated to humans and where key translational gaps remain.
- This is a very good point. As you commented, murine caspase-11 and human caspase-4/5 may differ in many aspects. Therefore, extrapolation of murine caspase-11 findings using animal models to humans as well as key possible translational gaps have been discussed in the “Conclusions and perspectives” section (Line 429-456).
7. Both NLRP3 and caspase-4/5/11 inflammasomes are discussed, but it is not clear whether noncanonical activation acts as an independent driver of pathology or mainly serves to amplify canonical NLRP3 signaling. Authors could clarify this, as it will would improve the mechanistic framework and help identify which pathway is the more promising therapeutic target.
- Thank you for your comments. Indeed, activation of the caspase-11 noncanonical inflammasome induces GSDMD pore formation, leading to K⁺ efflux through these pores, a critical trigger for NLRP3 inflammasome activation. Conversely, the NLRP3 inflammasome can promote caspase-11 noncanonical inflammasome activation via coordinated interactions with bacterial PAMPs during infection. These findings indicate that the caspase-11 noncanonical inflammasome functionally interplays with the NLRP3 canonical inflammasome in inflammatory and infectious contexts, rather than operating independently. Nevertheless, direct evidence for this functional interplay in diabetes and diabetes-associated complications remains lacking and requires further investigation. These are discussed in the “Conclusions and perspectives” section (Line 466-474).
8. At relevant place in manuscript, the manuscript could distinguish more explicitly between noncanonical inflammasome activation in metabolic endotoxemia (arising from microbial LPS leakage) and in sterile injury contexts (such as high glucose or ROS). Highlighting these differences would provide sharper mechanistic insight and point to areas where further research is needed.
- This is a good point and thank you for your suggestion. Unfortunately, this review discussed only 11 studies examining the involvement of murine caspase-11 and human caspase-4 noncanonical inflammasomes in diabetes and its complications. Also, the precise roles of these inflammasomes and their underlying molecular mechanisms remain largely unresolved. Consequently, clearly differentiating noncanonical inflammasome activation in the contexts of metabolic endotoxemia versus sterile injury is challenging and limited. Therefore, their roles and involvement were discussed according to disease types, without separating sterile from non-sterile injury contexts.

Round 2
Reviewer 3 Report
Comments and Suggestions for Authors
All comments has been addressed well